# Usefulness of Clinical and Laboratory Criteria for Diagnosing Autoimmune Liver Disease among Patients with Systemic Lupus Erythematosus: An Observational Study

**DOI:** 10.3390/jcm10173820

**Published:** 2021-08-26

**Authors:** Rebecca Heijke, Awais Ahmad, Martina Frodlund, Lina Wirestam, Örjan Dahlström, Charlotte Dahle, Stergios Kechagias, Christopher Sjöwall

**Affiliations:** 1Clinic of Internal Medicine, Region Jönköping County, SE-553 05 Jönköping, Sweden; rebecca.heijke@gmail.com; 2Department of Biomedical and Clinical Sciences, Division of Inflammation and Infection/Clinical Immunology, Linköping University, SE-581 85 Linköping, Sweden; awais.ahmad@regionostergotland.se (A.A.); charlotte.dahle@regionostergotland.se (C.D.); 3Department of Biomedical and Clinical Sciences, Division of Inflammation and Infection/Rheumatology, Linköping University, SE-581 85 Linköping, Sweden; martina.frodlund@regionostergotland.se (M.F.); lina.wirestam@liu.se (L.W.); 4Department of Behavioural Sciences and Learning, Swedish Institute for Disability Research, Linköping University, SE-581 85 Linköping, Sweden; orjan.dahlstrom@liu.se; 5Department of Health, Division of Diagnostics and Specialist Medicine/Gastroenterology and Hepatology, Medicine and Caring Sciences, Linköping University, SE-581 85 Linköping, Sweden; stergios.kechagias@liu.se

**Keywords:** abnormal liver function tests, autoimmune liver diseases, autoimmune hepatitis, hepatic involvement, liver biopsy, primary biliary cholangitis, systemic lupus erythematosus

## Abstract

Abnormal liver function tests are frequently observed during follow-up of patients with systemic lupus erythematosus (SLE) but data on co-existence with autoimmune liver diseases (AILD) are scarce. This retrospective study aimed to describe the prevalence of autoimmune hepatitis (AIH) and primary biliary cholangitis (PBC) among well-characterized subjects with SLE. We also evaluated whether the presence of autoantibodies to complement protein 1q (C1q) and/or ribosomal P protein (anti-ribP) are, directly or inversely, associated with AIH, as proposed in some reports. The number of screened patients was 287 (86% females), and all cases were included in a regional Swedish cohort. Each subject of the study population met the 1982 American College of Rheumatology classification criteria and/or the Fries’ diagnostic principle. By applying the simplified diagnostic AIH criteria combined with persistent transaminasemia, 40 (13.9%) cases reached at least “probable AIH”. However, merely 8 of these had been diagnosed with AIH (overall AIH prevalence 2.8%). Neither anti-C1q nor anti-ribP associated significantly with AIH. By applying the recent PBC guidelines, 6 (2.1%) cases were found, but only 3 of them had actually been diagnosed with PBC and one additional subject was not identified by the guidelines (overall PBC prevalence 1.4%). Compared to prevalence data from the general Swedish population, both AIH and PBC were highly overrepresented in our study population. The sensitivity of the diagnostic AIH criteria was impeccable but the specificity was less impressive, mainly due to positive ANA and hypergammaglobulinemia. Based on our findings, among subjects with SLE, the AIH criteria are less useful and liver biopsy combined with detection of other AILD-associated autoantibodies should be performed.

## 1. Introduction

Although involvement of joints, skin, mucous membranes, serosa and kidneys is common among patients with systemic lupus erythematosus (SLE) virtually any organ system may be affected. However, hepatic involvement has not been considered a primary organ manifestation in SLE as it is not included in any of the commonly used and recently updated classification criteria [1,2]. Nevertheless, the British Isles Lupus Assessment Group’s (BILAG) disease activity index includes ‘lupus hepatitis’ as a separate item but liver disease is not reflected in the more widely used SLE disease activity score 2000 (SLEDAI-2K) [1,3,4,5,6]. Still, abnormal liver function tests (LFTs) at any time-point are common in patients with SLE; reported numbers range from 9–60% depending on study population and limitations applied for abnormal values [7,8,9,10,11]. Potential causes of abnormal LFTs in SLE are numerous and include drug-induced liver injury (DILI), steatosis, viral hepatitis, vascular thrombosis and autoimmune liver disease (AILD). However, clinically significant AILD associated with SLE has been reported to be rare, and hepatic liver involvement appears to have a limited influence on mortality [11,12,13,14,15,16].

Observations of ’lupus-associated hepatitis’ or ’lupus hepatitis’ usually refer to an asymptomatic transaminasemia, consistent with SLE disease activity [9,17], which normalizes during glucocorticoid treatment [17]. Lupus hepatitis has been reported in 3–9% of patients [9,18,19]. The histopathological findings of lupus hepatitis are variable and non-specific, but mild portal inflammatory infiltrate, lobular necrosis and fatty infiltration are frequently found [9]. One study reported that intense deposits of complement protein 1q (C1q) were found in the majority of cases with lupus hepatitis, but unfortunately neither circulating C1q levels or anti-C1q antibodies—which often parallel SLE disease activity—were investigated [20,21,22,23]. Presence of autoantibodies to ribosomal P protein (anti-ribP) has been reported to associate with lupus hepatitis or autoimmune hepatitis (AIH), but contradictory results have also been published [17,18,24,25].

Whereas lupus hepatitis has been considered as a manifestation of SLE, AIH is regarded as a separate disease. However, the two conditions often share several features, including hypergammaglobulinemia, arthralgia and presence of antinuclear antibodies (ANA) [26]. Similarly to SLE, AIH also has a female predominance. The histopathology is characterized by progressive hepatocellular necrosis and inflammation, which untreated may lead to cirrhosis and end-stage liver disease. Although AIH prevalence data are uncertain, epidemiological studies from Scandinavia, Spain and New Zealand have estimated a prevalence of 12–25 per 100,000 inhabitants [27,28,29,30].

Data on co-existence of SLE and AIH are scarce [19]. Differential diagnostics may be challenging since elevated LFTs, hypergammaglobulinemia, a positive ANA test and response to glucocorticoid treatment are characteristic of both conditions. Efe et al. found that up to two thirds of SLE patients with abnormal LFTs fulfilled the simplified AIH criteria whereas only approximately 14% had histopathology compatible with AIH [7]. This illustrates that a liver biopsy is often necessary for a definitive diagnosis of AIH among patients with SLE [31].

Primary biliary cholangitis (PBC) constitutes another AILD with female predominance, which is chronic, often progresses, and can result in end-stage liver disease [32]. The prevalence of PBC in Sweden has been estimated to be approximately 15 per 100,000 inhabitants [33]. PBC is typically associated with the presence of anti-mitochondrial antibodies of M2 type (AMA-M2), which, in addition to persistently elevated serum alkaline phosphatase (ALP) and liver histology consistent with PBC, constitutes one of the three diagnostic criteria [34]. In addition, other subtypes of ANA, such as anti-speckled 100-kDa (Sp100), anti-promyelocytic leukemia protein (PML) and anti-glycoprotein 210-kDa (gp210), are also strongly associated with PBC [35]. Whereas the association of PBC with primary Sjögren’s syndrome (SS) is well documented [36,37], co-existence with SLE has been reported as rare [38]. In contrast, however, we recently showed that PBC-associated autoantibodies in SLE are relatively common [39].

The primary aim of the present study was to describe the prevalence of AILD in well-characterized Swedish SLE patients from a tertiary referral center. This was done with support from an experienced hepatologist (S.K.) using different grounds for the diagnoses of AIH and PBC. We further aimed to test whether the presence of anti-C1q and/or anti-ribP antibodies were associated with AIH.

## 2. Materials and Methods

### 2.1. Study Population

The study population consisted of 287 patients (248 women, 39 men) diagnosed with SLE (detailed in Table 1). All subjects had been included in the prospective and observational research program *Clinical Lupus Register in North-Eastern Gothia* (Swedish acronym ‘KLURING’) at the Rheumatology unit, University Hospital in Linköping [40]. Most (284 of 287 (99%)) patients met the Fries’ diagnostic principle and 243 of 287 (84.7%) fulfilled the 1982 American College of Rheumatology (ACR) classification criteria (mean number of fulfilled ACR criteria was 4.8, range 3–9) [5,41]. The study population corresponds to virtually all prevalent and incident adult SLE cases in the catchment area of Region Östergötland (approximately 365,000 adult inhabitants) between September 2008 and May 2020. The medical records of all patients were retrospectively reviewed with a focus on AILD.

### 2.2. Data Collection

AILD diagnoses, attributed by gastroenterologists among the 287 SLE cases, were retrieved from medical records, reviewed by a hepatologist (S.K.) and considered as ‘golden standard’. Furthermore, we applied the simplified diagnostic AIH criteria and the recent PBC guidelines from the American Association for the Study of Liver Diseases and examined their diagnostic performance in our study population [44,45]. In addition, we recorded the presence of secondary SS (diagnosis confirmed by a rheumatologist and defined according to classification criteria) and antiphospholipid syndrome (APS) [42,43]. Data on liver biopsies and liver imaging were also retrieved.

### 2.3. Laboratory Analyses

All patients had undergone continuous monitoring of liver enzyme values as was previously described [39]. Levels of IgG (normal range 6.7–15 g/L) and IgM (0.27–2.1 g/L) in plasma were recorded. IgG-ANA was detected by indirect immunofluorescence (IF) microscopy on fixed HEp-2 cells and anti-ribP antibodies were analyzed by FIDIS™ Connective profile, Solonium software v 1.7.1.0 (Theradiag, Croissy-Beaubourg, France) at the Clinical Immunology laboratory, University Hospital in Linköping [46,47]. Anti-C1q antibodies were analyzed by ELISA [48]. Hepatitis B surface antigen (HBsAg) and antibodies against hepatitis C virus (anti-HCV) were assessed by routine methods at the Clinical Microbiology laboratory, University Hospital in Linköping. Autoantibodies associated with AILD were analyzed as previously described [39].

### 2.4. Statistics

Associations between laboratory variables (categorical) and AILD were examined with Fisher’s exact test for significance and using Φ as a measure of association. *p*-values ≤0.05 were considered statistically significant. Statistical analyses were performed using the SPSS software version 26.0.0.0 (SPSS Inc., Chicago, IL, USA). Sensitivity (proportion of subjects correctly identified with AILD), specificity (proportion of subjects correctly identified without AILD), accuracy (proportion of correctly classified subjects), positive predictive value (PPV; proportion of AILD-classified subjects that are true AILD) and negative predictive value (NPV; proportion of non–AILD-classified subjects that are true non-AILD), were calculated, including 95% confidence intervals (CI) using the Wilson score method.

### 2.5. Ethical Approval

Oral and written informed consent were obtained from all participants. The study protocol was approved by the regional ethics review board in Linköping (Decision number M75–08/2008).

## 3. Results

Of the 287 screened patients, 182 (63.4%) had occasional or persistent elevations of aspartate aminotransferase (AST), alanine transaminase (ALT), alkaline phosphatase (ALP), and/or γ-glutamyl transferase (GGT) during follow-up. Less than 10% had elevated LFTs for >3 months. All subjects with confirmed AILD (*n* = 12) were found among the patients with elevated LFTs. In the subgroup with elevated LFTs (*n* = 182), 4.4% had a confirmed diagnosis of AIH and 2.2% of PBC.

As illustrated in Figure 1A, we applied the diagnostic AIH criteria to the study population [44]. The prevalence of AIH in the entire study population was 2.8% (*n* = 8); and establishment of the diagnosis included liver biopsy in 6 of 8 cases. The sensitivity and the NPV of the AIH criteria was high, but the specificity and PPV were lower (Table 2). Among the entire study population, 102 of 226 (45.1%) had hypergammaglobulinemia at least once. HBsAg and anti-HCV were absent in 77 of 102 (75.5%) and IF-ANA was positive (titer > 1:80) in 284 of 287 (99.0%) patients. According to the AIH criteria, 46 (16.0%) reached at least “probable AIH”, even without histopathological evaluation, whereof 40 cases had a history of elevated LFTs (AST and/or ALT). Of note, plasma IgG had not been measured in 61 subjects, which clearly limited the possibility of reaching “probable AIH”. However, a closer review of those reaching “probable AIH” revealed a different explanation than AIH in a majority of cases with elevated LFTs. Still, the presence of hypergammaglobulinemia (≥16 g/L) at any time-point during follow-up was significantly associated with a confirmed diagnosis of AIH (Φ = 0.16, *p* = 0.015).

Only 4 subjects had a confirmed diagnosis of PBC (1.4%), which included histopathological evaluation in 2 of 4. As shown in Figure 1B, we used the recent diagnostic guidelines for PBC by requiring elevation of ALP in combination with presence of typical PBC-associated antibodies (AMA-M2/Sp100/gp210) [45]. This procedure yielded 6 cases (2.1%). The specificity and the NPV of the guidelines for PBC were impressive, but the sensitivity and PPV among our study population were lower (Table 2).

Anti-C1q antibodies were detected in 60 of 260 subjects (23.1%), but the presence of anti-C1q did not associate significantly with AIH. Anti-ribP antibodies were detected in 17 of 247 patients (6.9%), but in none of those with a confirmed diagnosis of AIH. Altogether, 9 of 17 (52.9%) anti-ribP positive individuals had a history of elevated LFTs, which did not significantly differ from anti-ribP negative subjects. SLE with secondary SS did not associate with either AIH or PBC, although a non-significant trend was observed for the latter (Φ = 0.09, *p* = 0.16). APS was found in one patient with AIH (12.5%), but in none with PBC.

## 4. Discussion

Continuous supervision of LFTs is mandatory for patients using disease-modifying anti-rheumatic drugs (DMARDs), such as methotrexate, azathioprine, mycophenolate mofetil, leflunomide or cyclosporine. Abnormal LFTs in the absence of liver-related symptoms are very common during follow-up of patients with SLE. The cause may be obvious but is most often uncertain. Occasionally, the LFTs normalize without specific interventions or with a minor increase of the daily glucocorticoid dose. However, LFTs can also be persistently elevated, which often leads to both further investigations and interruption of a needed and well-functioning DMARD treatment. These cases may be challenging and often require interdisciplinary discussions between rheumatologists, hepatologists and immunologists.

In this study, we aimed to describe the prevalence of AILD among well-characterized SLE patients from a tertiary referral center. We used a previously described cohort in which we had the possibility to longitudinally monitor LFTs, liver imaging, biopsies, concomitant diagnoses, drugs as well as additional laboratory data, including results from autoantibody testing [39,40]. All data were discussed and evaluated by specialists in rheumatology (C.S.), hepatology (S.K.) and immunology (C.D.). The prevalence rates achieved in our study population for established AIH (2.8% among all, and 4.4% among those with elevated LFTs) and PBC (1.4% among all, and 2.2% among those with elevated LFTs) are in line with, or close to, what previously has been published. The recent review by González-Regueiro et al. mentions a prevalence of AIH of approximately 5–10% among SLE patients with abnormal LFTs [31]. For PBC, a lower prevalence (2.5–5%) has been observed [38].

To set these figures into a context, the AILD prevalence rates in our study population can be compared with prevalence data from the general Swedish population. Danielsson-Borssén et al. determined the point prevalence of AIH in 2009 to 17.3/100,000 inhabitants (0.17‰), which strongly contrasts to our finding of 2.8% in SLE [27]. Only older prevalence data from the time span 1973–1982 are available for PBC, when Danielsson et al. reported 15.1/100,000 inhabitants (0.15‰), which is considerably lower than the 1.4% found among patients with SLE herein [33]. Based on these findings, we conclude that both AIH and PBC are over-represented among patients with SLE. Similarly, AIH has been associated with an increased risk of developing systemic autoimmune diseases [49]. This is also in line with the overall empirical knowledge that “one autoimmune disease predisposes to another”. Interestingly, the recent years’ genetic advances have taught us that identical risk genes are shared by several different autoimmune conditions [50,51].

We also took the opportunity to challenge the simplified diagnostic AIH criteria and the recent guidelines for PBC diagnosis. The AIH criteria have previously been criticized for poor performance in SLE, especially with regard to low specificity [7]. However, to the best of our knowledge, their performance has not systematically been evaluated in a population of well-characterized SLE cases. Basically, our findings confirm the observation by Efe et al. and emphasize that liver biopsy is often needed for a definitive diagnosis of AIH among patients with SLE [31]. The PPV of reaching “probable AIH” herein was only 17%. However, plasma IgG ≥ 16 g/L associated significantly with confirmed AIH. This is of relevance as initially high levels of IgG have been associated with poor outcome in patients with combined SLE/AIH [52]. Although the sensitivity and PPV were mediocre, the recent diagnostic guidelines for PBC overall performed slightly better than the AIH criteria in our study population. This was mostly explained by the high diagnostic specificity for the PBC-associated ANA subtypes, compared to the non-specified ANA detected by IF microscopy, which is valid as one of the criteria for AIH.

In SLE, the high prevalence of ANA and hypergammaglobulinemia confers that the other diagnostic markers, i.e., SMA antibodies, antibodies against soluble liver antigen (SLA) or LKM, play a more important role for diagnosing AIH. These antibody specificities as well as PBC-specific ANA, such as Sp100, anti-PML and anti-gp210, in addition to AMA-M2, should therefore be included in the screening algorithm when AILD is suspected. However, in most of our cases reaching the level of ”probable AIH”, these autoantibody specificities had not been requested since their LFTs were not persistently elevated and the clinical suspicion of AILD had therefore not been raised. Furthermore, it is worth noting that for detection of SMA, a serum dilution that results in a cut-off corresponding to the 95th percentile of a healthy population should be used and must be evaluated by each laboratory. According to the diagnostic AIH criteria a serum dilution of 1:40 is recommended, but is actually based on experience from ”the old days” when the quality of the microscopes were much lower than today’s modern equipment [44,47].

Levels of the SLE-associated antibodies anti-C1q and anti-ribP are known to fluctuate over time [22,23,47]. An etiological role for anti-ribP in triggering both lupus hepatitis and AIH has been proposed [18,24,25]. Herein, we considered all patients once positive for anti-ribP or anti-C1q as positive and potential seroconversion over time was neglected. Still, we did not confirm the previously reported association between anti-ribP and AIH. However, according to a detailed review by Bessone et al., the association remains both uncertain and controversial [17]. This study has some limitations. It cannot be excluded that the actual prevalence of AILD in SLE is even greater than what we have estimated here [13]. Guided examination of AILD is usually driven by elevated LFTs, usually over a longer period of time, as the risk of progression to cirrhosis is associated with raised LFTs [53,54]. However, in our study population, none of the cases who reached “probable AIH” had persistently elevated (≥6 months) LFTs. Otherwise, the latter would normally have resulted in a liver biopsy. Nevertheless, from an international perspective, it is worthwhile underlining that the Nordic reference limits regarding AST and ALT are unusually high [55]. Thus, based on the liberal LFT reference intervals we apply in Sweden, it is not impossible that subclinical AILD to some extent may pass under the radar. Another limitation was the ethnic composition of the study population—almost 90% of the enrolled patients were of Caucasian origin. Ethnicity is indeed known to affect both disease severity and manifestations of SLE [56]. Thus, extrapolation of our results to other populations should be done with caution. In contrast, the study has several strengths. For instance, the fact that Swedish health care is universally available to all residents significantly reduces the risk of selection bias and ensures a high coverage of cases. The well-characterized cohort of SLE patients, longitudinally followed by a limited number of experienced rheumatologists at a single tertiary referral center, constitutes another advantage [40,47].

## 5. Conclusions

In conclusion, both AIH and PBC are overrepresented in SLE. When the diagnostic AIH criteria were applied, even higher numbers were achieved, but the use of these criteria cannot be recommended in patients with SLE. Instead, liver biopsy and detection of autoantibodies with higher diagnostic specificity for AILD could aid in the search (particularly for AIH) among individuals with SLE.

## Figures and Tables

**Figure 1 jcm-10-03820-f001:**
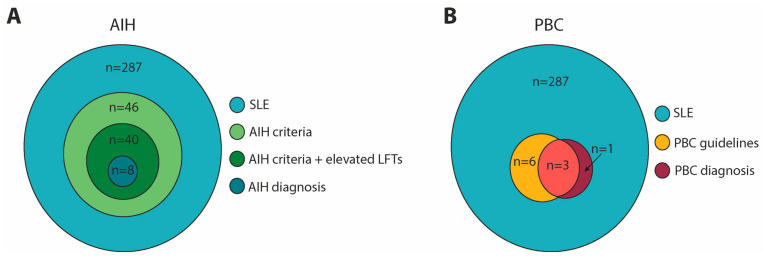
Diagrams illustrating the performance of the diagnostic AIH criteria (**A**) and PBC guidelines (**B**) among our study population. AIH = autoimmune hepatitis; SLE = systemic lupus erythematosus; PBC = primary biliary cholangitis; LFTs = liver function tests.

**Table 1 jcm-10-03820-t001:** Characteristics of the included patients with SLE. Patients with confirmed AILD are shown in separate columns.

Background Variables	Total, *n* = 287	AIH, *n* = 8	PBC, *n* = 4
Females, *n* (%)	248 (86.4)	7 (87.5)	3 (75.0)
Age at cohort inclusion, mean years (range)	49.0 (20–86)	47.8 (22–70)	53.2 (28–69)
SLE duration at cohort inclusion, mean years (range)	9.4 (0–45)	8.0 (0–19)	11.0 (1–25)
Caucasian ethnicity, *n* (%)	255 (88.8)	7 (87.5)	4 (100.0)
Ever smoker (former or current), *n* (%)	121 (42.1)	5 (62.5)	2 (50.0)
Disease variables			
Secondary Sjögren‘s syndrome (defined by classification ^#^), *n* (%)	65 (22.6)	2 (25.0)	2 (50.0)
Antiphospholipid syndrome (defined by classification ^§^), *n* (%)	56 (19.5)	1 (12.5)	0 (0)
Patients meeting ≥4 ACR-82 criteria, *n* (%)	243 (84.7)	6 (75.0)	3 (75.0)
Number of fulfilled ACR-82 criteria, mean (range)	4.8 (3–9)	4.8 (3–9)	3.8 (3–4)
Clinical phenotypes (ACR-82 defined), *n* (%)			
(1) Malar rash	114 (39.7)	4 (50.0)	0 (0)
(2) Discoid rash	43 (15.0)	0 (0)	0 (0)
(3) Photosensitivity	147 (51.2)	4 (50.0)	3 (75.0)
(4) Oral ulcers	34 (11.8)	1 (12.5)	1 (25.0)
(5) Arthritis	220 (76.7)	6 (75.0)	3 (75.0)
(6) Serositis	110 (38.3)	3 (37.5)	3 (75.0)
Pleuritis	106 (36.9)	3 (37.5)	2 (50.0)
Pericarditis	98 (34.1)	2 (25.0)	2 (50.0)
(7) Renal disorder	80 (27.9)	2 (25.0)	0 (0)
(8) Neurologic disorder	16 (5.6)	0 (0)	0 (0)
Seizures	15 (5.2)	0 (0)	0 (0)
Psychosis	3 (1.0)	0 (0)	0 (0)
(9) Hematologic disorder	174 (60.6)	5 (62.5)	0 (0)
Hemolytic anemia	12 (4.2)	0 (0)	0 (0)
Leukocytopenia	86 (30.0)	2 (25.0)	0 (0)
Lymphopenia	112 (39.0)	4 (50.0)	0 (0)
Thrombocytopenia	31 (10.8)	0 (0)	0 (0)
(10) Immunological disorder	152 (53.0)	4 (50.0)	1 (25.0)
Anti-dsDNA antibody (anti-dsDNA)	139 (48.4)	4 (50.0)	1 (25.0)
Anti-Smith antibody (anti-Sm)	19 (6.6)	1 (12.5)	0 (0)
(11) Antinuclear antibody (IF-ANA) *	284 (99.0)	8 (100)	4 (100)

* Positive by immunofluorescence (IF) microscopy. ^#^ According to Vitali C, et al. [42] ^§^ According to Miyakis S, et al. [43].

**Table 2 jcm-10-03820-t002:** Performance of the simplified diagnostic AIH criteria and the recent PBC guidelines [44,45] in the study population to identify SLE patients with confirmed AILD. Cut-off for the AIH criteria was set at ≥6 points (representing “probable AIH”). 95% confidence intervals are shown in parentheses.

	AIH	PBC
Sensitivity	1.00 (0.60–1.00)	0.75 (0.28–0.97)
Specificity	0.86 (0.82–0.90)	0.99 (0.97–1.00)
Accuracy	0.87 (0.82–0.90)	0.99 (0.96–1.00)
PPV	0.17 (0.09–0.31)	0.50 (0.19–0.81)
NPV	1.00 (0.98–1.00)	1.00 (0.98–1.00)

PPV = Positive Predictive Value; NPV = Negative Predictive Value; AIH = autoimmune hepatitis; PBC = primary biliary cholangitis.

## Data Availability

Not applicable.

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
