# Peer review of "Usefulness of Clinical and Laboratory Criteria for Diagnosing Autoimmune Liver Disease among Patients with Systemic Lupus Erythematosus: An Observational Study"

_jcm, 2021, doi:10.3390/jcm10173820_

Round 1
Reviewer 1 Report
In this study the Authors investigated the presence of AILD in SLE patients. The title is quite informative, the aim are clear. Methods are well devised and Results support the conclusions.
Minor comments
- The number of screened patients must be moved in Results section (both in Abstract and main text)
- A more specific paragraph about study limitations would be welcomed
- The title is quite misleading as it does not deal about an actual “interdisciplinary challenge”. Maybe a clear reference to the observational nature of the study and prevalence of AILD in SLE patients would be better
Author Response
Thank you very much for a thorough review with constructive comments on our manuscript.
The number of screened patients are now clearly stated in Abstract as well as in the Results section.
As requested, a more specific paragraph about study limitations has been included at the end of the Discussion section.
The title of the manuscript has been changed as suggested (Usefulness of clinical and laboratory criteria for diagnosing autoimmune liver disease among patients with systemic lupus erythematosus: An observational study).
Reviewer 2 Report
This is a very nice paper reviewing abnormal LFTs in a well described cohort with SLE. It confirms a high prevalence of AIH and PBC and a lack of specificity with standard diagnostic approaches, along with no association with anti-c1q and anti-ribP.
There are a few very minor typographical suggestions:
Line 74 'diagnosing' should be 'diagnosis'
Line 92 'reported rare' should be 'reported as rare'
Line 110 'with focus' should be 'with a focus'
Line 223 'with in SLE' should be 'with SLE'
Line 227 'genetic conquests' should be something like 'genetic advances'
Line 261 'in an international' should be 'from an international'
Author Response
Thank you very much for a thorough review with constructive comments on our manuscript.
The text has been corrected in accordance with the typographical suggestions.